# Maize Transcription Factor *ZmHsf28* Positively Regulates Plant Drought Tolerance

**DOI:** 10.3390/ijms24098079

**Published:** 2023-04-29

**Authors:** Lijun Liu, Yuhan Zhang, Chen Tang, Qinqin Shen, Jingye Fu, Qiang Wang

**Affiliations:** State Key Laboratory of Crop Gene Exploration and Utilization in Southwest China, College of Agronomy, Sichuan Agricultural University, Chengdu 611130, China; xshanghe@163.com (L.L.); yuhanzhang994@163.com (Y.Z.); tangchen4723@163.com (C.T.); shenqin2016@sicau.edu.cn (Q.S.)

**Keywords:** maize, drought tolerance, jasmonate, ABA, transcription factor

## Abstract

Identification of central genes governing plant drought tolerance is fundamental to molecular breeding and crop improvement. Here, maize transcription factor *ZmHsf28* is identified as a positive regulator of plant drought responses. *ZmHsf28* exhibited inducible gene expression in response to drought and other abiotic stresses. Overexpression of *ZmHsf28* diminished drought effects in Arabidopsis and maize. Gene silencing of *ZmHsf28* via the technology of virus-induced gene silencing (VIGS) impaired maize drought tolerance. Overexpression of *ZmHsf28* increased jasmonate (JA) and abscisic acid (ABA) production in transgenic maize and Arabidopsis by more than two times compared to wild-type plants under drought conditions, while it decreased reactive oxygen species (ROS) accumulation and elevated stomatal sensitivity significantly. Transcriptomic analysis revealed extensive gene regulation by *ZmHsf28* with upregulation of JA and ABA biosynthesis genes, ROS scavenging genes, and other drought related genes. ABA treatment promoted *ZmHsf28* regulation of downstream target genes. Specifically, electrophoretic mobility shift assays (EMSA) and yeast one-hybrid (Y1H) assay indicated that *ZmHsf28* directly bound to the target gene promoters to regulate their gene expression. Taken together, our work provided new and solid evidence that *ZmHsf28* improves drought tolerance both in the monocot maize and the dicot Arabidopsis through the implication of JA and ABA signaling and other signaling pathways, shedding light on molecular breeding for drought tolerance in maize and other crops.

## 1. Introduction

Drought is the main adverse factor to crop growth and threatens agricultural production worldwide. With global climate change, increasing population, and a limited fresh water supply, improvement of drought tolerance in crop production has emerged as an urgent problem [1].

Heat shock factor (Hsf) is the transcription factor (TF) family existed in the eukaryotes including higher plants, and is widely involved in plant stress responses, such as to heat, salinity, drought, and immunity [2]. Plant Hsfs are divided into three sub-groups: HsfA, B, and C. In tomato, HsfA1 regulated DREB2A, HsfA2, and HsfA7a to mediate thermotolerance [3,4,5]. In pepper, wheat, and tall fescue, HsfAs promoted heat tolerance through regulating various stress response genes [6,7,8,9,10]. AtHsfA7b positively regulated salinity tolerance [11]. OsHsfA3 elevated abscisic acid (ABA) biosynthesis and polyamine production, resulting in improved drought tolerance in transgenic Arabidopsis [12]. AtHsfB1 and AtHsfB2b were demonstrated to be necessary for the expression of heat shock protein genes in Arabidopsis [13]. The mutant of *OsHsfC1b* exhibited lower tolerance to salt and osmotic stress, indicating the positive role in salinity tolerance and growth [14]. In addition, some Hsfs also play roles in plant immunity. AtHsfB1 and AtHsfB2b regulated PDF1.2 expression and related disease resistance [15]. OsHsfB4d regulated *OsHsp18.0-CI* expression to enhance rice bacterial blight resistance [16].

Plant hormones play important roles in stress responses. ABA promotes stomatal closure, improves water absorption and utilization, and osmotic metabolite accumulation to provide drought tolerance [17,18,19]. ABA is synthesized with serial catalysis by some key enzymes, such as zeaxanthin epoxidase (ZEP), 9-cis-epoxycarotenoid dioxygenase (NCED), and aldehyde oxidase (AAO) [20,21,22,23]. The ABA receptors PYR/PYLs/RCAR perceive ABA and recruit the repressor PP2C, leading to de-association of the PP2C-SnRK2 complex and the release of SnRK2. SnRK2s are activated by auto-phosphorylation and phosphorylate downstream target proteins, resulting in further signaling transduction and gene expression regulation, and, eventually, physiological/biochemical changes in response to drought stress [18,24,25,26]. Increasing endogenous ABA production is a reasonable strategy to enhance plant drought tolerance. In wheat, TaFDL2-1A promotes ABA biosynthesis, signaling, and ROS scavenging to elevate drought resistance [27]. In maize, ZmbZIP33, ZmbZIP4, and ZmPTF1 increase ABA biosynthesis gene expression under drought stress, leading to ABA accumulation and activation of ABA signaling [22,28]. The elevated endogenous ABA signaling activates specific gene expression of *ZmSLAC1* in guard cells, promoting stomatal closure and drought resistance [29].

Jasmonate (JA) is widely involved in plant growth and defense including drought stress response. JA is synthesized from α-linolenic acid through the key intermediate 12-oxo-phytodienoic acid (OPDA) via sequential catalysis by lipoxygenase (LOX), allene oxide cyclase (AOS), allene oxide cyclase (AOC), OPDA reductase 3 (OPR3), and JAR1 [30,31]. The active JA-Ile is recognized by the JA receptor SCFCOI1, leading to the repressor JAZ degradation and releasing the key regulator MYC2 to activate the downstream JA response [32]. Overexpression of *CmLOX10* in Arabidopsis enhanced drought tolerance and JA accumulation and signaling [33]. In the drought tolerant variety of chickpea, the JA biosynthesis genes responded in the earlier stage in response to drought, promoting high accumulation of JA in roots [34]. Overexpression of *JAR1* in Arabidopsis increased JA-Ile content, leading to drought adaptation, including changed stomatal density and aperture, as well as reactive oxygen species (ROS) detoxification [35].

In maize, a few Hsfs have been identified, including four HsfAs and two HsfBs with involvement in heat, salinity, and drought tolerance [36,37,38,39,40,41]. Nevertheless, most studies focused on HsfA and B members, and HsfCs have not been investigated substantially. Here, to explore the function of maize Hsfs in drought responses, we identified *ZmHsf28*, a C-subgroup Hsf, as the positive regulator in drought tolerance through involvement in JA and ABA signaling pathways to elevate plant drought tolerance. *ZmHsf28* exhibited inducible gene expression in response to multiple abiotic stresses. Overexpression of *ZmHsf28* increased drought resilience in transgenic Arabidopsis and maize, while gene silencing of *ZmHsf28* via VIGS diminished drought tolerance in maize. Lower ROS accumulation and higher JA and ABA production were detected in Arabidopsis and maize with *ZmHsf28* overexpression. Transcriptomic analysis and further promoter regulation assays revealed that *ZmHsf28* extensively regulated downstream target genes to play a positive role in drought resistance.

## 2. Results

### 2.1. Identification of ZmHsf28 in Response to Drought Treatment

Based on the previous report, *ZmHsf28* was detected with strong induction in response to drought [42,43]. Through phylogenetic analysis, *ZmHsf28* was identified as the member of HsfCs in 31 maize Hsfs and showed the close relationship with rice HsfC proteins but not with Arabidopsis Hsfs (Appendix A). *ZmHsf28* contains the conserved domains of Hsf proteins (Appendix A). Natural drought and PEG6000 significantly induced *ZmHsf28* expression, indicating involvement in drought response. Slight upregulation of *ZmHsf28* was detected under treatment with NaCl or heat. In addition, methyl jasmonate (MeJA) treatment drastically induced *ZmHsf28* expression, and moderate upregulation was detected by exogenous ABA, suggesting the role of *ZmHsf28* in JA and ABA signaling (Figure 1A–F). *ZmHsf28* was observed to be localized in the nucleus (Figure 1G), suggesting the function as the TF.

### 2.2. Overexpression of ZmHsf28 Elevated Drought Tolerance in Arabidopsis

To explore the potential function of *ZmHsf28* to affect drought resistance in plants, we overexpressed *ZmHsf28* in the dicot Arabidopsis thaliana. Overexpression of *ZmHsf28* did not affect seedling growth negatively but significantly elevated drought tolerance (Figure 2A,B). The leaves of ZmHsf28OE Arabidopsis exhibited lower water loss and higher sensitivity of stomatal closure to ABA and PEG treatments (Figure 2C–E). Lower ROS accumulation was also observed in ZmHsf28OE Arabidopsis under drought treatment through DAB and NBT staining (Figure 2F,G), as well as higher anti-oxidant enzyme activities and lower MDA accumulation (Figure 2H–L). In addition, no significant difference was observed for JA and ABA contents between the ZmHsf28OE and wild-type (WT) Arabidopsis under the normal growth condition (CK); however, twice as much JA and ABA were produced in Arabidopsis with *ZmHsf28* overexpression than in WT under drought treatment (Figure 2M,N).

### 2.3. ZmHsf28 Regulated Drought Tolerance in Maize

To further elucidate the biological function of *ZmHsf28* in maize, VIGS was conducted, and successful silencing of *ZmHsf28* was confirmed by qRT-PCR analysis (Figure 3A). The silencing lines showed severe wilting, and the survival rate was much lower than the control after drought (Figure 3B,C). The stable maize overexpression lines was generated to investigate the function of *ZmHsf28*. In the seedling stage, ZmHsf28OE lines showed no growth difference under the well-watered condition. After natural drought, ZmHsf28OE and WT plants showed symptoms of drought stress, but only moderate wilting was observed for the overexpression lines (Figure 3D). The survival rate of ZmHsf28OE lines was much higher than WT after re-watering (Figure 3E). At the reproductive stage, no significant difference was detected for most agronomic traits including the kernel yield under the normal growth condition (Figure 3F,G). After drought treatment, the overexpression lines showed better growth with higher values for many agronomic traits, especially the kernel yield (Figure 3F–J). Taken together, overexpression of *ZmHsf28* did not affect maize growth and yield negatively but elevated drought tolerance to rescue growth and yield at the seedling and reproductive stages.

### 2.4. Overexpression of ZmHsf28 Promoted Physiological Adaptation to Drought

Physiological changes were observed in maize with an overexpression of *ZmHsf28*. Under the normal growth condition, no apparent difference in stomatal movement was observed between the WT and ZmHsf28OE lines. When treated with ABA or PEG6000, ZmHsf28OE lines exhibited higher sensitivity with a much lower stomatal aperture and conductance, indicating stronger water retention in response to drought (Figure 4A–D). A lower abundance of ROS was also accumulated in ZmHsf28OE lines after drought treatment through DAB and NBT staining (Figure 4E), indicating more efficient ROS scavenging.

JA and ABA contents were measured, and no significant difference was detected in WT and ZmHsf28OE lines under the normal growth conditions, with low contents of JA and ABA in all tested plants (Figure 4F–I). After drought treatment, JA and ABA biosynthesis was highly induced but much higher production was detected in ZmHsf28OE lines than WT with different times (5.5 times for ABA, 2.9 times for JA, 8.6 times for JA-Ile, and 2.2 times for OPDA) (Figure 4F–I). These results indicate that *ZmHsf28* strongly promotes JA and ABA biosynthesis in response to drought but not under the normal condition, suggesting some regulatory mechanism to fine-tune *ZmHsf28* function in response to drought.

### 2.5. Overexpression of ZmHs28 Reprogrammed Extensive Gene Expression in Maize

Transcriptional changes in ZmHsf28OE lines and WT were compared by RNA-seq. Either under the normal growth condition (CK) or drought, overexpression of *ZmHsf28* extensively affected gene expression and led to transcriptional reprogramming. Under CK, 6947 differentially expressed genes (DEGs) were detected, including 2856 upregulated and 4091 downregulated DEGs (Figure 5A, Appendix A). Under drought treatment, 8104 DEGs were revealed (Figure 5A, Appendix A). By comparing CK and drought treatment, 2796 common DEGs were detected in both treatments (Figure 5B). GO analysis indicated that these common DEGs were involved in abundant biological processes, such as the response to stress, hormone response, and oxidative stress response (Figure 5C, Appendix A). Specific DEGs were also detected in either drought treatment or CK. GO analysis indicated that specific DEGs in CK were mainly enriched in organic acid and carboxylic acid metabolic process and some primary metabolism. In contrast, specific DEGs in drought were mainly enriched in transmembrane transporter activity, the oxoacid metabolic process, and the regulation of hormone levels (Figure 5D, Appendix A).

JA and ABA biosynthesis genes were analyzed using transcriptomic data (Appendix A). Many JA biosynthesis genes were significantly induced by drought and accumulated more in ZmHsf28OE lines, consistent with the higher JA production under drought stress. Meanwhile, JAZ proteins exhibited lower gene expression in ZmHsf28OE lines, suggesting the activation of JA signaling and downstream response for drought resistance (Figure 5E). In addition, most ABA biosynthesis genes did not show apparent expression difference between ZmHsf28OE and WT plants in CK, but rather accumulated many more transcripts in ZmHsf28OE lines under drought treatment (Figure 5F). These results validated the higher ABA accumulation in ZmHsf28OE lines under drought conditions. Moreover, ABA receptor genes ZmPYL2 and ZmPYL8 were highly induced by drought with higher transcript accumulation in ZmHsf28OE lines. SnRK2s exhibited differential responses, although all of them were induced by drought, and only SnRK2.2 and SnRK2.5 showed higher expression in ZmHsf28OE lines. PP2Cs exhibited differential gene expression patterns between ZmHsf28OE and WT plants, while most of them were induced by drought in either groups (Figure 5F).

Some anti-oxidant enzymes involved in ROS scavenging were also upregulated, with enhanced gene expression in ZmHsf28OE lines. These data explained the physiological phenotype of lower ROS production in ZmHsf28OE lines in response to drought (Figure 5G). Additionally, some drought response marker genes, such as ZmRAB18, ZmRD20, ZmRD21, and ZmDREB2A, were observed with higher expression in ZmHsf28OE lines (Figure 5H), further demonstrating the regulatory function of *ZmHsf28* in drought tolerance. Moreover, abundant genes in other signaling and biological processes were also detected with differential expression by *ZmHsf28* overexpression under drought stress, such as a number of TFs, kinases, aquaporins, VQ motif proteins, and late embryogenesis abundant proteins, suggesting the extensive regulatory effect of *ZmHsf28* in drought response (Appendix A).

Here, qRT-PCR analysis was conducted to validate the RNA-seq data, and all tested genes did not exhibit significant expression differences between WT and ZmHsf28OE lines under CK. In contrast, much higher expression was revealed in ZmHsf28OE lines with drought stress, consistent with the transcriptomic analysis (Figure 6). We noticed that some drought response genes, including JA and ABA biosynthesis genes, were not upregulated by *ZmHsf28* under CK but were drastically elevated under drought, suggesting some mechanisms to enhance *ZmHsf28* activity to increase plant drought tolerance.

### 2.6. ABA Treatment Promoted ZmHsf28 Regulation on Target Genes

Overexpression of *ZmHsf28* did not change the growth phenotype under CK but increased resistance to drought (Figure 2 and Figure 3). Correspondingly, some genes were specifically upregulated in ZmHsf28OE lines (Figure 5 and Figure 6), which might be attributed to some specific mechanisms under drought stress, such as ABA signaling. Hence, *ZmHsf28* was transiently overexpressed in maize protoplasts with or without ABA treatment, and five potential target genes were detected, including JA and ABA biosynthesis genes, anti-oxidant genes, and drought response marker genes (Figure 7A–E). Consistent with the results in seedlings, these genes did not show significant upregulation or were only slightly induced by *ZmHsf28* transient overexpression compared to the control. Upon treatment with ABA, significant induction was observed, and much higher expression was detected with *ZmHsf28* transient overexpression, indicating that *ZmHsf28* regulation to downstream genes was enhanced by ABA, consistent with the results in seedling under drought treatment.

We further analyzed the activation of *ZmHsf28* on these target gene promoters by promoter-luciferase assays. As shown in Figure 7F–J, *ZmHsf28* did not significantly activate or only slightly upregulated these gene promoters without ABA treatment, consistent with the corresponding gene expression analysis. Upon addition of ABA, strong activation was observed by *ZmHsf28* for all five tested promoters. Further deletion analysis revealed the active regions of these gene promoters. Upon deleting these active regions, the promoter activity was diminished, and no activation was detected by *ZmHsf28*. Specifically, the active promoter region of ZmRD21 was determined as the promoter fragment at −2043 to −1170 bp upstream of the start codon (Figure 7F). For ZmCAT1, the fragment of −1400 to −600 bp was identified as the active region (Figure 7G). For ZmZEP1 and ZmAOS1, the active regions were identified at −1080 to −620 bp and −1600 to −800 bp, respectively (Figure 7H,I). ZmAAO4 was regulated by *ZmHsf28* at −1850 to −1300 bp (Figure 7J).

### 2.7. ZmHsf28 Directly Bound to Target Gene Promoters

We further analyzed the binding of *ZmHsf28* to target gene promoters. The promoter fragments used for promoter–luciferase assay were also used for Y1H assay. As indicated in Figure 8A, Y1H assays demonstrated the direct binding of *ZmHsf28* to gene promoters for all five tested genes. Abundant heat shock elements (HSEs) were uncovered in these gene promoters and might be bound by *ZmHsf28* (Figure 7F–J, Appendix A). The promoter fragments containing HSEs, as revealed by deletion analysis, were synthesized for EMSA assay. The specific HSEs of five tested genes were used with the following positions: −1564 bp for *ZmRD21*, −1290 bp for *ZmCAT1*, −935 bp for *ZmZEP1*, −1540 bp for *ZmAOS1*, and −1732 bp for *ZmAAO4* (Appendix A). EMSA further demonstrated that *ZmHsf28* directly bound to these promoter fragments with specific HSEs (Figure 8B), further validating the regulatory function of *ZmHsf28* on these genes.

## 3. Discussion

There are strict control systems for growth and resistance balance in plants to allocate energy and resource to either party precisely in response to diverse environments [44,45]. Here, *ZmHsf28* was identified as the positive regulator of drought response. Overexpression of *ZmHsf28* did not affect plant growth but significantly elevated tolerance to drought and rescued the yield, exhibiting the potential in crop breeding as an elite gene.

We noticed that drought treatment or ABA application promoted *ZmHsf28* regulation of downstream target genes. ABA accumulation is induced in plants by drought treatment, and further downstream signaling transduction mainly relies on phosphorylation [46,47]. In Arabidopsis, AREB1/ABF2, AREB2/ABF4, and ABF3 were activated by SnRK2s in response to osmotic stress [48]. OsbZIP23 was phosphorylated by OsSAPK2, leading to promoted *OsNCED4* expression and ABA biosynthesis, which eventually enhanced drought resistance [49]. Phosphorylation of AtHSFA4 at Ser309 by MPKs resulted in elevated salinity and oxidative stress tolerance [50,51]. *ZmHsf28* might be phosphorylated by ABA signaling under drought stress to promote downstream gene expression to elevate drought tolerance. Future investigations should be conducted to analyze putative phosphorylation of *ZmHsf28* and related mechanisms.

JA signaling also plays important roles to regulate drought response independently or through cooperating with ABA [52,53,54]. Drought treatment induced JA and ABA accumulation in maize with earlier induction for JA production [55]. We also detected high induction of ABA and JA related compounds in maize and Arabidopsis in this study. Exogenous application of ABA or JA improved the tolerance to PEG treatment in pearl millet through elevating ROS scavenging [56]. Overexpression of *VaNCED1* increased ABA accumulation in grape, as well as JA biosynthesis, leading to enhanced drought resistance [57]. Production of JA-Ile induced by water stress is necessary for ABA biosynthesis in Arabidopsis roots [58]. Overexpression of *SlMYC2* in tomato promoted ROS production in guard cells and accumulation of JA and ABA in leaves to drive stomatal closure and increase drought tolerance [59]. Overexpression of *ZmHsf28* did not significantly affect JA and ABA biosynthesis under the normal condition but activated JA and ABA production upon drought stress, subsequently increasing drought tolerance. We observed the upregulation of JA and ABA biosynthetic genes by overexpression of *ZmHsf28* in maize plants, as well as promoter activation in maize protoplasts with transient overexpression of *ZmHsf28*, indicating that *ZmHsf28* regulates plant drought tolerance through JA and ABA biosynthesis and signaling.

The function of Hsfs was originally linked to heat stress response, and, later, broader roles were revealed in various stress responses and growth [60]. Most studies focus on HsfA and B sub-families in plants and comprehensively characterize their functions and related molecular mechanisms [61,62]. However, only a few C-type Hsfs have been investigated. FaHsfC1b played positive roles in heat stress tolerance, and TaHsfC2a-B was also identified as a positive regulator of heat protection [63,64,65]. OsHsfC1b was involved in salt and osmotic stress response and also played roles in plant growth [14]. Here, we identified *ZmHsf28*, the C-type Hsf, as the positive regulator in maize drought response. *ZmHsf28* can directly bind to the HSEs of target genes and activate the corresponding expression. *ZmHsf28* extensively regulated downstream genes to mitigate drought effects, including activation of JA and ABA biosynthesis. As far as our knowledge goes, this is the first report about the C-type Hsf function in the regulation of drought tolerance, and our findings indicate that HsfCs also play important roles in plant stress responses.

Higher plants evolve some universal mechanism to adapt to drought stress, such as ABA-mediated stomatal closure, ROS scavenging, and osmoprotectant accumulation [66,67,68,69]. Such processes are fine-tuned to maintain the balance between growth and drought resistance. Overexpression of some resistant genes led to elevated drought tolerance but greatly impaired growth and yield, which is not acceptable for agricultural production. Overexpression of *OsDREB1A* or *OsPYL5* enhanced drought resistance but caused stunted growth and decreased yield [70,71,72]. TINY positively regulated drought response but inhibited BR-mediated growth [73]. However, some genes were also reported to increase drought tolerance without growth penalty or even better growth. For example, ZmPIF1 was involved in ABA-mediated stomatal closure to elevate drought resistance, as well as increasing tillers and panicle numbers to improve rice yield [74]. ARGOS improved drought resistance in Arabidopsis and maize, and led to greater grain yield [75]. *ZmHsf28* increased drought tolerance in maize and the dicot plant Arabidopsis through regulating some universal mechanisms, including ABA and JA biosynthesis, ROS scavenging, and stomatal closure, but did not affect plant growth negatively, suggesting the great potential for crop improvement against drought stress.

In summary, *ZmHsf28* exhibited inducible gene expression in response to drought, and overexpression of *ZmHsf28* did not cause negative effects on plant growth but, rather, improved drought resilience in maize and Arabidopsis through promoting stomatal closure and ROS scavenging. *ZmHsf28* also enhanced JA and ABA accumulation under drought, and ABA significantly promoted regulation of *ZmHsf28* to downstream target genes, highlighting its role in JA and ABA signaling pathways and crosstalk between plant hormones. Future work could be conducted to reveal the detailed regulatory mechanism of *ZmHsf28*, such as protein phosphorylation and de-phosphorylation, protein interaction, and potential protein modification.

## 4. Materials and Methods

### 4.1. Plant Materials

Maize inbred lines B73, Mo17, and KN558, and *ZmHsf28* overexpression (ZmHsf28OE) lines (with KN5585 background) were germinated and grown in the growth room with 16 h light and 8 h dark at 28 °C. *A. thaliana* (Col-0) and ZmHsf28OE Arabidopsis lines (Col-0 background) were germinated and grown with 14 h light and 10 h dark at 22 °C.

### 4.2. Treatments, Gene Expression Analysis and Cloning

Mo17 seedlings (2 weeks) were treated with 20% PEG6000 or heat (42 °C) for 24 h or 250 mM NaCl, 100 μM MeJA, or 50 μM ABA for 48 h, respectively. Natural drought was performed without watering for 15 d. Above-ground tissues were collected for further gene expression analysis. Total RNA was extracted using the TRNzol reagent (TianGen, Beijing, China), and cDNA was synthesized with M-MLV reverse transcriptase (Takara, Tokyo, Japan). qRT-PCR was performed using the SYBR GREEN master mix (Gangchi Bio, Hangzhou, China) on a StepOne Plus Real-Time PCR System (Applied Biosystems, Waltham, MA, USA) [76]. All experiments were conducted with at least three biological replicates. *ZmHsf28* (Zm00001d046299) was cloned from Mo17 leaves with 24 h PEG treatment. Gene promoters were amplified from gDNA of Mo17. Primers used were listed in Appendix A.

### 4.3. Transgenic Plants

KN5585 was used for generation of ZmHsf28OE lines. *ZmHsf28* was subcloned into pCAMBIA3301 with the control of the *Ubiquitin* promoter for maize transformation. *A. thaliana* Col-0 was used to generate the transgenic lines of *ZmHsf28* under the control of the 35S promoter [77]. The transgenic plants at T3 generation were used for further analysis.

### 4.4. Virus-Induced Gene Silencing (VIGS)

VIGS of *ZmHsf28* was conducted as described previously [78]. The maize seeds with VIGS were germinated and grown until the three-leaf stage. The seedlings were treated with natural drought for 10 d and re-watered for 3d, and the survival rate was determined. The silencing efficiency was analyzed through qRT-PCR at the seventh day of drought treatment. All experiments were conducted with at least three biological replicates.

### 4.5. Drought Treatments of Transgenic Plants

KN5585 and ZmHsf28OE maize lines (2 weeks) were grown in the same pots and treated by natural drought for 20 d and re-watered for 5 d, then the survival rate was calculated. The above-ground tissues were collected at day 10 for further analysis ,including RNA-seq, ROS-related experiments, and plant hormone measurement.

KN5585 and ZmHsf28OE maize plants at the flowering stage were treated by natural drought for 10 d and normal watering was restored until harvesting. The soil water content was measured to monitor the degree of drought and dropped from 35% on the 1st day to 13% on the 10th day during drought treatment. Agronomic traits were recorded, and 20 plants were used for each line as biological replicates.

ZmHsf28OE Arabidopsis and Col-0 plants (4 weeks) were treated without watering for 10 d and were re-watered for 3 d, and then the survival rate was determined. The above-ground tissues were collected at the seventh day for further analysis.

### 4.6. Water Loss Rate

The above-ground tissues of Arabidopsis plants (4 weeks) were detached and kept at room temperature for weighing at different time points to assess water loss. Five seedling were used for each line, and three biological replicates were performed.

### 4.7. ROS Detection, Anti-Oxidant Enzyme Activity and MDA Measurement

Maize seedlings (2 weeks) were treated with natural drought for 10 d, and the leaves were collected for DAB and NBT staining, as described previously [79]. Arabidopsis plants treated with natural drought for 7 d were collected for ROS detection as above. Anti-oxidant enzyme activity and MDA content were measured as described previously [80], using the samples with the same treatment as that for ROS detection. All experiments were conducted with at least three biological replicates.

### 4.8. Measurement of Stomatal Aperture and Conductance

The maize (2 weeks) and Arabidopsis plants (4 weeks) were used to observe stomatal movement as describe previously [81,82]. Here, 15% PEG4000 or 10 μM ABA was used to treat leaves for 2 h, and stomatal movement was observed with a microscope. A total of 20 stomata were observed to measure stomatal aperture and conductance for each line. Stomatal aperture was measured using the ImageJ V1.8.0 software [83]. Stomatal conductance was measured by an LI-6800 portable photosynthesis system (LI-COR, Lincoln, NE, USA). KN5585 and ZmHsf28OE maize plants (3 weeks) were treated with 15% PEG6000 or 50 μM ABA, and stomatal conductance was measured at 0, 2, 4, 8, and 12 h during treatment.

### 4.9. Plant Hormone Measurement

The leaves of maize seedlings with natural drought for 10 d were used for ABA and JA derivative measurements. For Arabidopsis, four-week-old plants with natural drought for seven days were collected for ABA and JA measurements. Plant hormone detection and quantification were performed on the AB SciexQTRAP 6500 LC-MS/MS platform. Three biological replicates were analyzed for each line.

### 4.10. Protoplast Preparation and Transient Overexpression

Maize leaf protoplasts were isolated as described previously [76]. *ZmHsf28* was subcloned into pCAMBIA2300 to generate the corresponding construct. Plasmids were transfected into protoplasts and kept in the dark for 12 h, and continuous incubation with the addition of 50 μM ABA was conducted for 6 h. The empty vector was transfected parallel as the negative control. Total RNA extraction and cDNA synthesis were performed as described above. Downstream gene expression was determined by qRT-PCR analysis. Experiments were carried out for at least three biological replicates.

### 4.11. Subcellular Localization

*ZmHsf28* was subcloned into pCAMBIA-GFP without the stop codon and further transformed into maize protoplasts for subcellular localization analysis, as described previously [79]. The H2B (p35S-H2B-mCherry) was used as the nuclear localization marker. Fluorescence was observed under a confocal microscope (Leica, Wetzlar, Germany).

### 4.12. Promoter-Luciferase Assays

Promoters of target genes were cloned based on the sequence information in MaizeGDB and subcloned into pGreenⅡ 0800-LUC. *ZmHsf28* was ligated into pCAMBIA2300 as described above. The plasmids of promoters and transcription factors were co-transfected into maize leaf protoplasts with different combinations, as above. The empty vector was used as the control. The promoter activity was determined and represented with the ratio of firefly luciferase and Renilla luciferase (LUC_Firefly_/LUC_Renilla_), as described previously [84]. The deletion analysis of promoter was conducted as described previously [85]. Each experiment was performed with 6–8 biological replicates. Primers were listed in Appendix A.

### 4.13. Yeast One-Hybrid (Y1H) Assays

The Y1H assay was conducted as described previously [86]. Specifically, *ZmHsf28* was subcloned into pGADT-7 to generate AD-ZmHsf28. The promoter fragments of target genes were connected into pAbAi. The interaction was determined by yeast growth selection on SD/-Ura/-Leu and SD/-Ura/-Leu/AbA. The empty pGADT-7 was transformed as the negative control.

### 4.14. Electrophoresis Mobility Shift Assay (EMSA)

*ZmHsf28* was subcloned into pET28a for recombinant expression in the *Escherichia coli* BL21(DE3) strain. The recombinant protein was purified using the Ni-NTA beads (Smart-Lifesciences, Changzhou, China). Promoter fragments of target genes containing specific HSE motifs were synthesized with biotin labeling (Sango, Shanghai, China). Unlabeled probes were used for competition. EMSA was conducted with a chemiluminescence EMSA kit (Beyotime, Shanghai, China), as described previously [85]. Probes were listed in Appendix A.

### 4.15. RNA-Seq Analysis

RNA-seq analysis was conducted using the samples of maize and Arabidopsis with drought treatment as described above. Five individual plants for each lines were collected with three replicates. Total RNA was extracted, and cDNA was synthesized. cDNA library sequencing and data filtering were conducted on an Illumina NovaSeq 6000 by Novogene (Beijing, China). Specifically, transcriptome libraries were prepared using a TruSeqTM RNA sample preparation kit from Illumina (San Diego, CA, USA). mRNA was isolated with polyA selection by oligo(dT) beads and fragmented. cDNA synthesis, end repair, A-base addition, and ligation of the Illumina-indexed adaptors were performed according to Illumina’s protocol. Paired-end libraries were sequenced by Illumina NovaSeq 6000 sequencing (150 bp × 2). Differentially expressed genes (DEGs) were analyzed by DESeq2, and GO and KEGG analyses were conducted by GSEA (www.gsea-msigdb.org/gsea/index.jsp (accessed on 15 March 2022)) [87,88].

### 4.16. Sequence Alignment, Phylogenetic Analysis and Cis-Element Prediction

Hsf protein sequences in maize, rice, and Arabidopsis were obtained from PlantTFDB v5.0 (planttfdb.gao-lab.org/ (accessed on 1 March 2020)). Sequence alignment was performed on MEGA 11 (64-bit) (www.megasoftware.net/ (accessed on 1 March 2020)) and a phylogenetic tree was constructed on iTOL v6 (itol.embl.de/ (accessed on 1 March 2020)). Prediction of cis-element was conducted on PlantPAN 3.0 (plantpan.itps.ncku.edu.tw/promoter.php (accessed on 10 May 2020)) and MEME 5.5.2 (meme-suite.org/meme/tools/meme (accessed on 10 May 2020)). The genes and accession IDs used for phylogenetic analysis were listed in Appendix A.

### 4.17. Statistical Analysis

All experiments were conducted with at least three biological replicates, and data were collected and analyzed to determine significant difference through Student’s *t*-test or ANOVA with Fisher’s LSD test using IBM SPSS statistics 20.

## Figures and Tables

**Figure 1 ijms-24-08079-f001:**
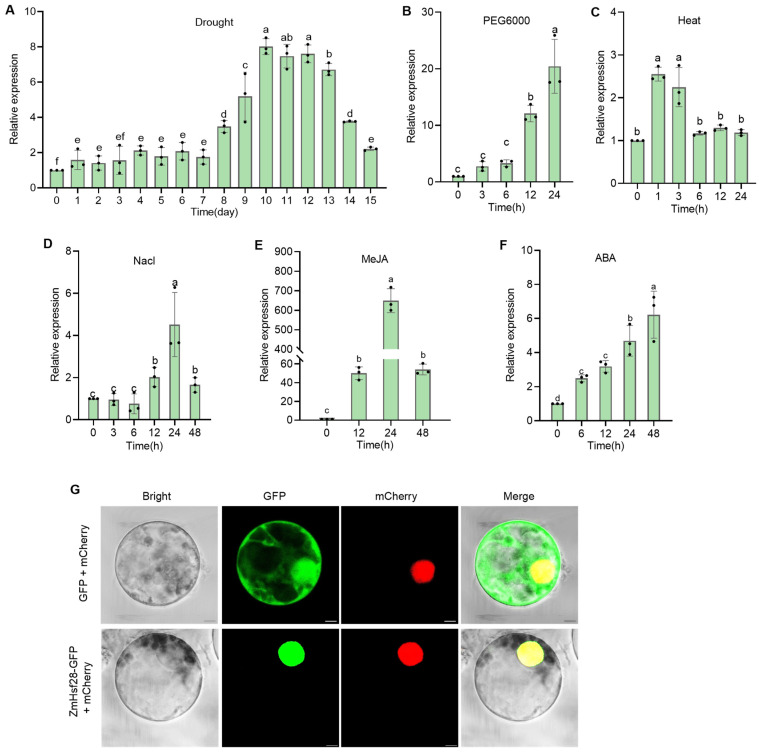
Inducible gene expression and subcellular localization of *ZmHsf28*. (**A**–**F**) qRT-PCR analysis of *ZmHsf28* gene expression in response to natural drought (**A**), PEG6000 (**B**), heat (**C**), NaCl (**D**), MeJA (**E**), and ABA (**F**). Different lowercase letters indicate the significant difference (one-way ANOVA followed by Fisher’s LSD test, *p* < 0.05). Error bars indicate mean ± SE. (**G**) Subcellular localization of *ZmHsf28* in maize protoplasts. The cell structures in green indicate the localization of GFP fusion proteins. The red spots indicate the nuclear localization of mCherry fusion proteins. The yellow spots indicate the merged subcellular localization of GFP and mCherry fusion proteins. H2B-mCherry was used as the nuclear marker. Bars, 20 μm.

**Figure 2 ijms-24-08079-f002:**
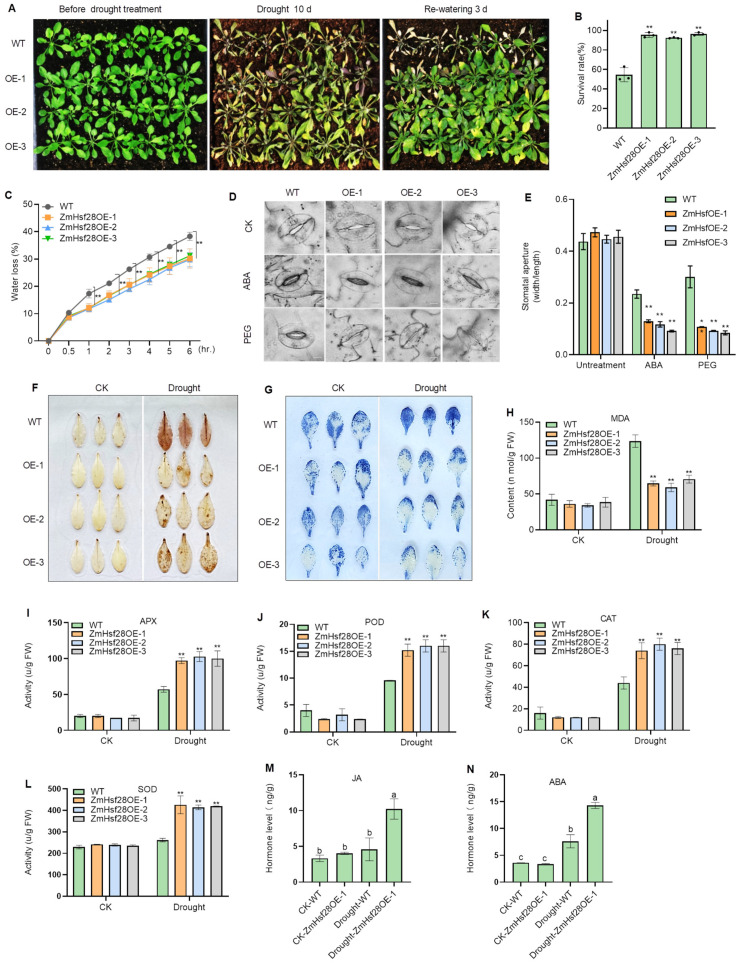
Overexpression of *ZmHsf28* elevated drought tolerance in Arabidopsis. (**A**) Arabidopsis plants with *ZmHsf28* overexpression (ZmHsf28OE-1, OE-2, OE-3) under drought treatment and re-watering. (**B**) Survival rate of Arabidopsis plants with *ZmHsf28* overexpression under drought treatment and re-watering. (**C**) Water loss rate of *ZmHsf28* overexpression in Arabidopsis plants and WT. (**D**) Stomata of *ZmHsf28* overexpression in Arabidopsis lines and WT in response to ABA or PEG treatment. CK is the control with the normal growth condition. Bars, 20 μm. (**E**) Stomatal aperture of *ZmHsf28* overexpression in Arabidopsis lines and WT with treatments of CK, ABA, or PEG. (**F**,**G**) DAB (**F**) and NBT (**G**) staining for Arabidopsis leaves of *ZmHsf28* overexpression lines and WT under the normal growth condition and drought treatment. (**H**) MDA contents in *ZmHsf28* overexpression in Arabidopsis lines (ZmHsf28OE-1, -2 and -3) and WT under drought treatment or normal condition (CK). (**I**–**L**) Anti-oxidant enzyme activities of ROS scavenging in *ZmHsf28* overexpression in Arabidopsis plants under drought treatment and CK, including APX (**I**), POD (**J**), CAT (**K**), and SOD (**L**). WT Arabidopsis plants were used as the control. (**M**,**N**) Contents of JA (**M**) and ABA (**N**) for Arabidopsis seedlings of *ZmHsf28* overexpression lines (ZmHsf28OE) and WT under CK and drought treatment. Asterisks indicate the significant difference (Student’s *t*-test, * *p* < 0.05, ** *p* < 0.01). Different lowercase letters indicate the significant difference (one-way ANOVA followed by Fisher’s LSD test, *p* < 0.05). Error bars indicate mean ± SE.

**Figure 3 ijms-24-08079-f003:**
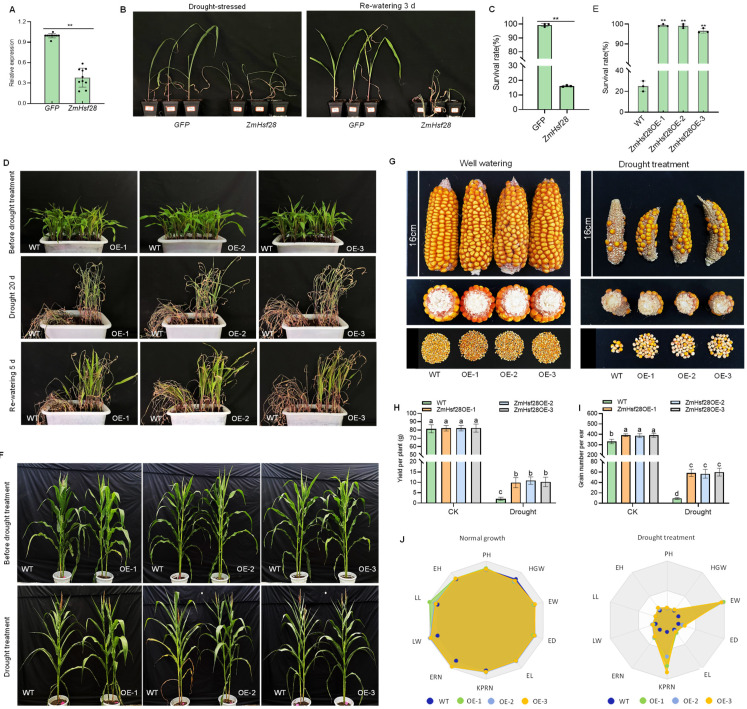
Overexpression of *ZmHsf28* elevated drought tolerance in maize. (**A**) Silencing efficiency of *ZmHsf28* by VIGS. (**B**) Maize seedlings with *ZmHsf28* silencing by VIGS after drought treatment and re-watering. (**C**) Survival rate of maize seedlings with *ZmHsf28* silencing after drought and re-watering. GFP was the control. (**D**) Maize seedlings with *ZmHsf28* overexpression for drought treatment and re-watering. Three overexpression lines (OE-1, -2, -3) and WT were treated. (**E**) Survival rate of *ZmHsf28* overexpression in maize plants with drought and re-watering. (**F**) Maize plants with *ZmHsf28* overexpression and WT at the flowering stage for drought treatment. (**G**) Ears and grains of *ZmHsf28* overexpression in maize plants and WT which were watered well and given drought treatment. (**H**,**I**) Yield per plant (**H**) and grain number per ear (**I**) of *ZmHsf28* overexpression in maize plants and WT which were watered well and given drought treatment. (**J**) Radar chart analysis for agronomic traits of *ZmHsf28* overexpression in maize plants and WT with normal growth and drought treatment. PH, plant height; LL, leaf length; LW, leaf width; EH, ear height; KPRN, number of kernels per row; EL, ear length; ED, ear diameter; EW, ear weight; ERN, number of ear row; HGW, hundred grain weight. Asterisks indicate the significant difference (Student’s *t*-test, ** *p* < 0.01). Different lowercase letters indicate the significant difference (one-way ANOVA followed by Fisher’s LSD test, *p* < 0.05). Error bars indicate mean ± SE.

**Figure 4 ijms-24-08079-f004:**
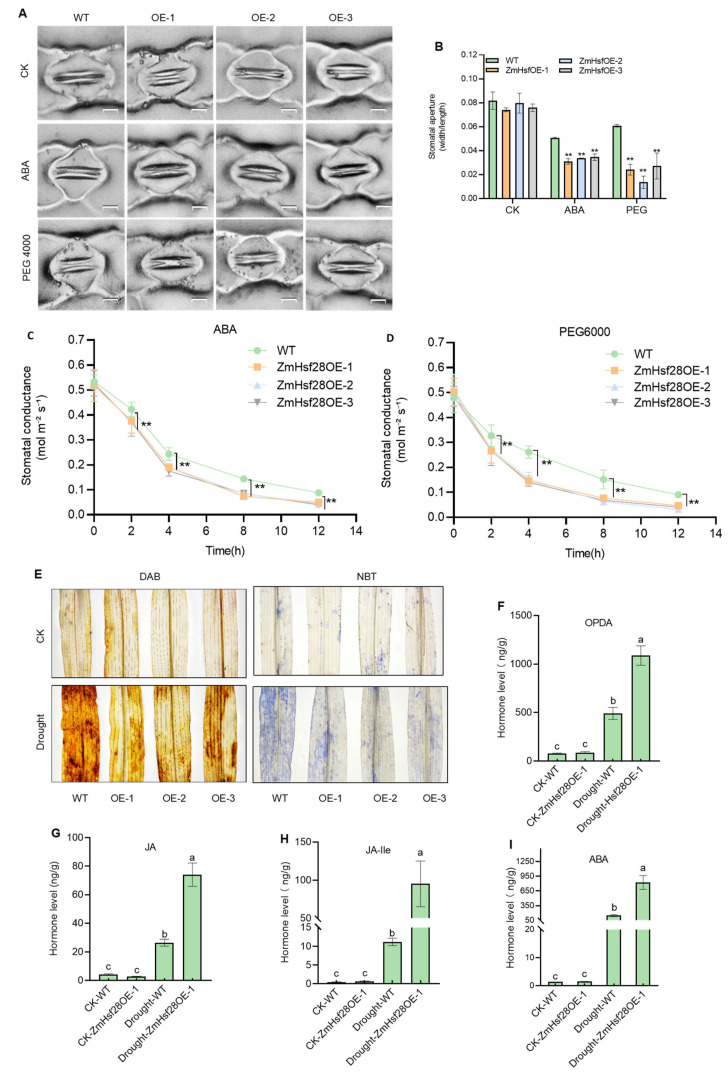
Physiological changes in maize plants with *ZmHsf28* overexpression. (**A**) Stomata of *ZmHsf28* overexpression in maize lines (ZmHsf28OE) and WT (KN5585) in response to ABA or PEG treatment. CK is the control. Bars, 20 μm. (**B**,**C**) Stomatal aperture (**B**) and conductance (**C**,**D**) of ZmHsf28OE and WT with treatments of CK, ABA, or PEG. (**E**) DAB and NBT staining for maize leaves of ZmHsf28OE and WT under CK and drought treatment. (**F**–**I**) Contents of plant hormones and related compounds for maize seedlings of ZmHsf28OE and WT under CK and drought treatment including OPDA (**F**), JA (**G**), JA-Ile (**H**) and ABA (**I**). Asterisks indicate the significant difference (Student’s *t*-test, ** *p* < 0.01). Different lowercase letters indicate the significant difference (one-way ANOVA followed by Fisher’s LSD test, *p* < 0.05). Error bars indicate mean ± SE (*n* = 3).

**Figure 5 ijms-24-08079-f005:**
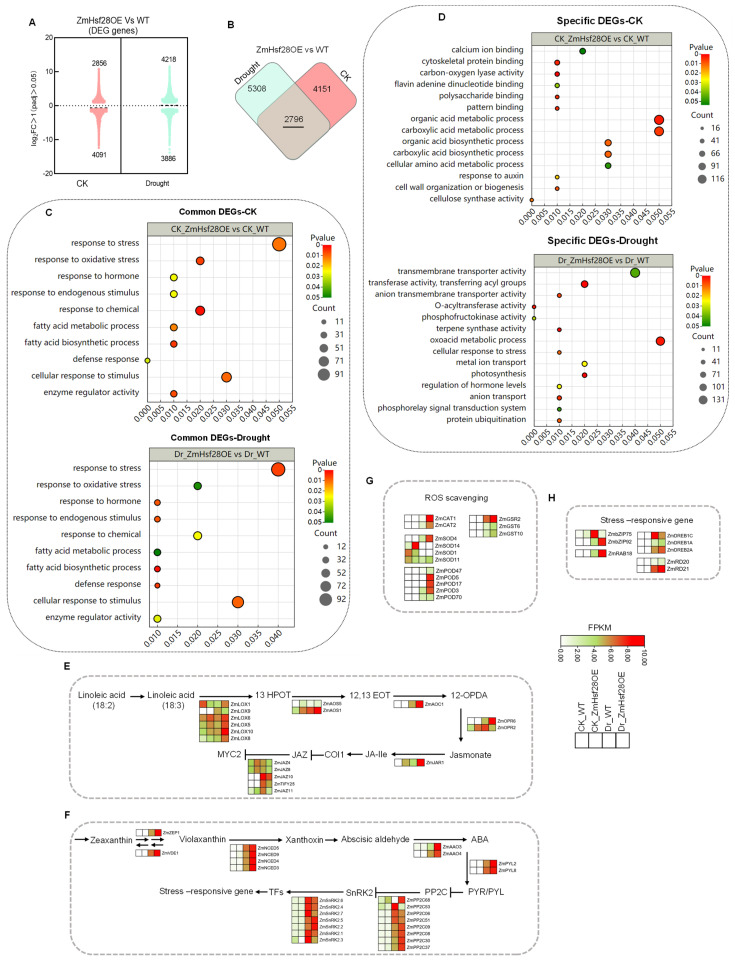
RNA-seq analysis of *ZmHsf28* overexpression in maize lines. (**A**) Violin plot of DEGs in *ZmHsf28* overexpression lines (ZmHsf28OE-1) and WT (KN5585) under the normal growth conditions (CK) and drought treatment. Log2FC indicates the fold change in upregulation or downregulation. (**B**) Venn diagram of common and specific DEGs between ZmHsf28OE and WT under CK and drought treatment (Dr). (**C**) GO enrichment analysis of common DEGs with CK and drought treatment. (**D**) GO enrichment analysis of specific DEGs with CK or drought treatment. (**E**–**H**) Heat map of DEGs involved in JA (**E**) and ABA (**F**) biosynthesis and signaling, ROS scavenging (**G**), and some stress-responsive genes (**H**). The colors indicate FPKM values.

**Figure 6 ijms-24-08079-f006:**
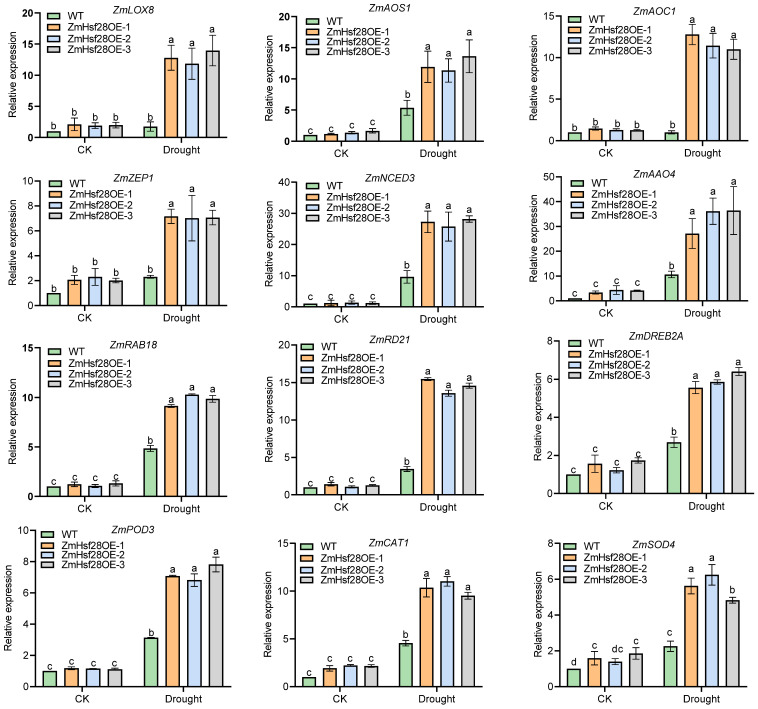
Gene expression validation of RNA-seq by qRT-PCR analysis. Maize genes involved in JA and ABA biosynthesis, ROS scavenging, and drought-responsive marker genes were analyzed in response to drought treatment or normal condition (CK) in *ZmHsf28* overexpression in maize lines (ZmHsf28OE) and WT (KN5585). Different lowercase letters indicate the significant difference (one-way ANOVA followed by Fisher’s LSD test, *p* < 0.05). Error bars indicate mean ± SE.

**Figure 7 ijms-24-08079-f007:**
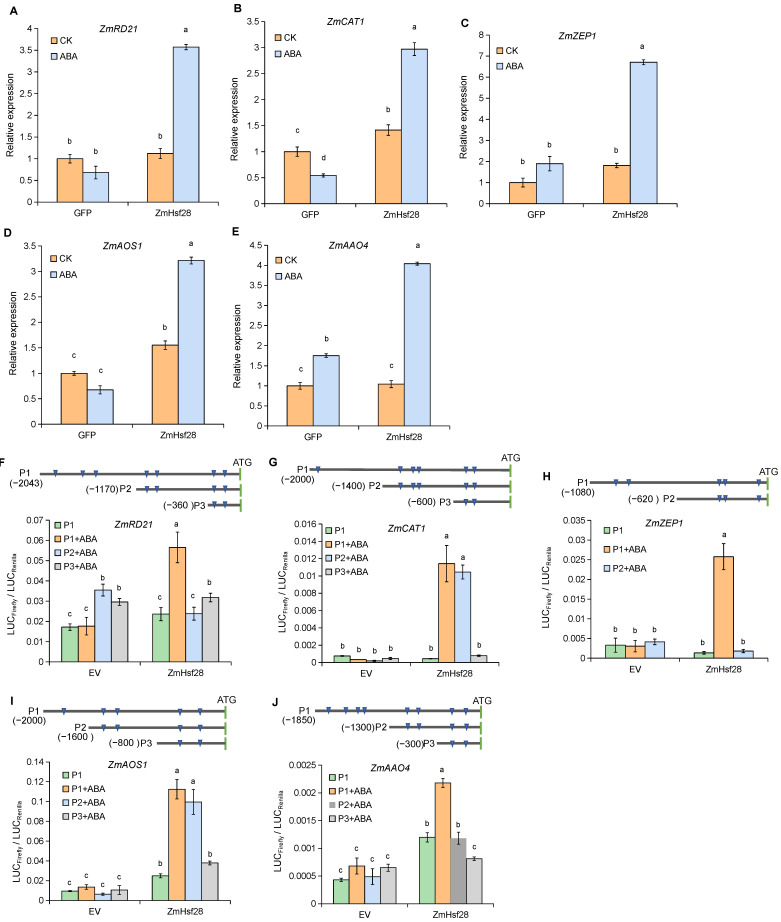
ABA treatment activated *ZmHsf28* regulation on downstream gene expression. (**A**–**E**) qRT-PCR analysis of downstream gene expression in maize protoplasts with transient overexpression of *ZmHsf28* plus ABA treatment. The empty vector with the GFP was used as the control. No ABA was added for CK. (**F**–**J**) Promoter–luciferase assays of *ZmHsf28* with target gene promoters. Promoter fragments were labeled with the sites upstream of the start codon, and HSEs were indicated with the triangles. *ZmHsf28* was transiently overexpressed in maize protoplasts with or without ABA treatment. The empty vector (EV) was used as the control. Promoter activity was indicated with LUC_firefly_/LUC_Renilla_. Different lowercase letters indicate the significant difference (one-way ANOVA followed by Fisher’s LSD test, *p* < 0.05). Error bars indicate mean ± SE (*n* = 3).

**Figure 8 ijms-24-08079-f008:**
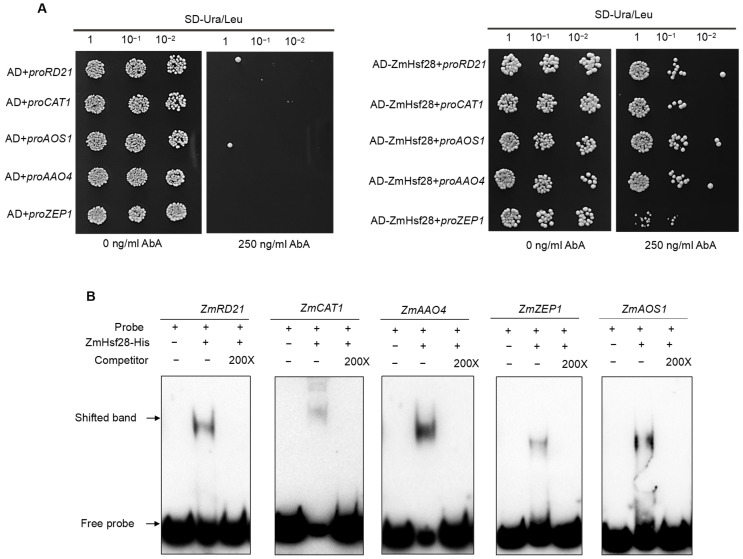
*ZmHsf28* binds to target gene promoters. (**A**) Y1H assays of *ZmHsf28* with target gene promoters. (**B**) EMSA of *ZmHsf28* with promoter fragments containing HSEs. Competitors were the cold probes without biotin labeling.

## Data Availability

Not applicable.

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
