# Peer review of "Maize Transcription Factor ZmHsf28 Positively Regulates Plant Drought Tolerance"

_ijms, 2023, doi:10.3390/ijms24098079_

Round 1
Reviewer 1 Report
This work is interesting, which is a significant advancement over existing knowledge, but it needs substantial improvements before further consideration for publication.
1. The abstract should contain some quantitative information.
2. Aims and objectives, please clearly indicate the main points undertaken in the aims and objective section before the material and method section.
3. 4.15 RNA-seq analysis; Write the detailed procedure of sequencing and library construction;
4. 4.17 Statistical analysis; Write details such as replications and software used for statistical analysis.
5. The abbreviation used must be explained on their first appearance, or provide a separate list of abbreviations
6. Please support your statements in the discussion section with relevant references and compare the results with the latest related studies.
There are several typographical and grammatical mistakes that should be corrected.
Reviewer 2 Report
the manuscript is significantly improved and can be accepted for publication but before final decision I am interested to know as author (s) overexpressed ZmHsf28 in arabidopsis what about arabidopsis hsf28 homologous genes. did author perform anay Analysis related to zmhsf28 similarities with arabidopsis hsf genes? please indicate.
in abstract author(s) mentioned VIGS in maize but i didn't see any results or figure showing phenotype of claim please check this.
Reviewer 3 Report
The manuscript addresses an important issue of maize drought tolerance. The study dissects the significant role of the transcription factor ZmHsf28 employing a laborious array of current molecular methods and tools. The experimental design is well organized, and the materials and methods are clearly presented. However, the manuscript requires extensive editing and reformat to present thoroughly the laborious amount of work employed to discern the role of the ZmHsf28 transcription factor to drought. Following are some indicative suggestions that the authors should elaborate to improve the manuscript.
1. In the abstract
a. ln 8 “Identification of resistant genes is crucial for molecular breeding to improve crop drought tolerance” could be rephrased to “Identification of central genes governing plant drought tolerance is fundamental to molecular breeding and crop improvement”. Ln 9 “regulator in plant drought response” should be “regulator of plant drought responses”. In ln 10 “to drought treatment and other abiotic stresses or treatments” should be rephrased to “to drought and other abiotic stresses”. Also, ln 11 “increased drought tolerance” should be rephrased to “enhanced drought tolerance”.
b. Ln 13-15 “Overexpression of ZmHsf28…” should be rephrased to “Overexpression of ZmHsf28 increased jasmonate (JA) and abscisic acid (ABA) production in maize and Arabidopsis under drought, while decreased ROS accumulation and stomatal sensitivity”.
c. Ln 17-18 “regulation on downstream target genes” should be rephrased to “regulation of downstream target genes” and the “ZmHsf28 directly bound…” should be rephrased to “Specifically electrophoretic mobility shift assays (EMSA) indicated that ZmHsf28 directly binds to the target gene promoter, regulating its expression “.
d. Ln 19-22 could be rephrased to “Taken together, our work provided new and solid evidence that ZmHsf28 improved the drought tolerance both in the monocot maize and the dicot Arabidopsis through the implication of JA and ABA signaling pathways, shedding light on molecular breeding for drought tolerance in maize and other crops”.
2. The Introduction section requires reform and rewriting to improve the article’s readership, providing orderly all relevant information. Thus, I would suggest that the paragraph of ln 58-74 to be moved to the top of the Introduction, following the introductory paragraph after ln 29. Using an introductory sentence regarding the HSFs (Heat Shock Factors) family and their roles in stress responses, followed by ln 58-74.
Then using a connection sentence regarding the role of plant hormones, place the paragraph of ln 30-45.
At the end of the Introduction in ln 75-76 describe briefly how the study explored meticulously the role of ZmHsf28 to drought stress under natural conditions as well under drought-induced treatments such as PEG. And how this role was verified in maize and Arabidopsis plants using overexpression systems and gene silencing (VIGS) and transcriptomics analyses. Also indicate the assessment of drought related responses including ROS scavenging enzyme activities, antioxidants content, plant hormones, stomatal sensitivity and agronomic parameters that were assessed in all systems to thoroughly dissect the role of ZmHsf28.
3. The authors are encouraged to carefully check the manuscript for English grammar and syntax errors (i.e., ln 417 “biological replicated” should be “biological replicates”, ln 440 “Plant hormone detection and quantified” should be “Plant hormone detection and quantification” etc.).
4. The term ‘enhanced drought tolerance” should not be constantly used as there is no clear measurement or comparison among different values of drought tolerance. Instead, the authors could use interchangeably the terms “diminished drought effects”, “increased drought resilience” and “increased plant robustness in drought”.
5. The conclusion, last paragraph in the Discussion section should be reformed to clearly present the valuable outcomes of the study. Specifically, the expression of the ZmHsf28 transcription factor is evident in wild type maize plants under drought stress. However, the overexpression of ZmHsf28 in both maize and Arabidopsis plants indicated an improved drought resilience, highlighting its role in signaling pathways and crosstalk with plant hormones. Also, the exogenous application of ABA on overexpressing ZmHsf28 plants/cell cultures resulted in the overexpression of protecting antioxidant gene/enzyme systems, while in control conditions overexpression of ZmHsf28 solely, did not induce downstream genes. Thus, unraveling the significant interplay of ZmHsf28 and plant hormone signaling pathways in response to drought stress.
The authors are encouraged to carefully check the manuscript for English grammar and syntax errors.
Round 2
Reviewer 1 Report
The authors addressed my suggestions proposed in the first round of revision. The manuscript can be accepted for publication.
Author Response
The authors addressed my suggestions proposed in the first round of revision. The manuscript can be accepted for publication.
Reply: Thanks for your work.